# OpenReview forum: "Malice in Agentland: Down the Rabbit Hole of Backdoors in the AI Supply Chain"
_ICLR.cc/2026/Conference — Submitted to ICLR 2026_

### Official Review · Reviewer_mhKC · 2025-10-31

**Soundness:** 2
**Presentation:** 2
**Contribution:** 2
**Rating:** 2
**Confidence:** 4

**Summary:**

This paper identifies a critical and previously underexplored vulnerability within the AI supply chain: the fine-tuning of AI agents on their own interaction data introduces a potent vector for backdoor attacks. The authors compellingly demonstrate that an adversary can poison the data collection pipeline at multiple points—directly in training traces, via the operational environment (e.g., web pages), or through a pre-backdoored base model—to implant a hidden trigger.

**Strengths:**

1. The figures in the paper are presented with exceptional clarity.

2. The paper is written in an accessible and easy-to-understand manner.

**Weaknesses:**

I have several concerns regarding the experimental section of this paper:

1. While the paper proposes a novel backdoor attack method, it only compares its performance against zero-shot prompting and SFT. It would be more compelling to include comparisons with other state-of-the-art backdoor attack methods to properly situate its contribution.

2. The scale of the datasets used appears limited. The test sets contain only 115, 50, and 165 test tasks, respectively. Employing larger-scale datasets would strengthen the reliability and generalizability of the findings.

3. The paper seems to lack ablation studies. Conducting such experiments is crucial to validate the contribution and necessity of each component within the proposed framework.

**Questions:**

The current experiments are conducted solely on 7B and 8B parameter LLMs. Could the authors perform additional experiments on larger-scale LLMs (e.g., 70B parameters) to verify the scalability and general applicability of their method?

---

> ### Author Response · Authors · 2025-11-26
>
> We thank Reviewer mhKC for the feedback.  We address concerns and questions below:
>
> **Use of SFT vs RL:** Please see the general response.
>
> **Benchmarks:** We selected these benchmarks as they are ubiquitous and standard in the agentic literature [7, 8]. Regarding the robustness, we would like to point out that all of our results include tight confidence intervals, confirming the significance of our findings despite the sizes of the benchmarks. Further, the fact that results generalize across domains strengthens their credibility, as pointed out by reviewer n4Sw.
>
> **Ablations:**  Please see the general response.
>
> **Model Scale:** Please see the general response.
>
> [7] Zhou, S., Xu, F. F., Zhu, H., Zhou, X., Lo, R., Sridhar, A., ... & Neubig, G. (2024). WEBARENA: A REALISTIC WEB ENVIRONMENT FOR BUILDING AUTONOMOUS AGENTS. In 12th International Conference on Learning Representations, ICLR 2024.
>
> [8] Yao, S., Shinn, N., Razavi, P., & Narasimhan, K. (2024). $\tau $-bench: A Benchmark for Tool-Agent-User Interaction in Real-World Domains. arXiv preprint arXiv:2406.12045.

---

### Official Review · Reviewer_n4Sw · 2025-11-01

**Soundness:** 2
**Presentation:** 2
**Contribution:** 2
**Rating:** 2
**Confidence:** 3

**Summary:**

This paper investigates the security vulnerabilities of LLM-based agents, particularly focusing on how LLM agents can be exploited or manipulated in realistic task environments. The authors build a framework that systematically explores prompt injection, environment manipulation, and multi-agent collusion as potential threat vectors. Using both simulated and real-world benchmarks, they demonstrate that these attacks can lead to unintended behaviors, such as resource misuse, task derailment, and data exfiltration across popular LLM agent architectures (e.g., AutoGPT, BabyAGI, and LangChain-based agents). The paper also proposes defense strategies, including input filtering, anomaly detection, and weight auditing. The experiments highlight both the feasibility of such attacks and the insufficiency of current safety mechanisms in agentic systems.

**Strengths:**

1. Timely and important topic: The security of LLM agents is an emerging area, and this paper addresses it comprehensively by considering different threat models and defenses accordingly.
2. The evaluation on different agent benchmarks and task types increases the robustness and generalizability of the results.

**Weaknesses:**

1. My main concern is that, while the paper frames three main threat models, these categories closely parallel existing backdoor and data poisoning paradigms in traditional and foundation models. Prior literature has already established strong conceptual and empirical analyses of such attacks in both supervised learning and LLM contexts [1, 2, 3, 4, 5, 6]. This work mainly extends existing threat models to LLM agents but does not clearly articulate what unique challenges arise from the agentic architecture, e.g., sequential decision-making, tool interaction, or memory persistence.
2. There is no ablation on agent components, thus it is still unclear how much each component (memory, planning, or tool use) contributes to the overall vulnerability. A finer-grained analysis could give us deeper insights.

[1] Gu et al., Badnets: Identifying vulnerabilities in the machine learning model supply chain.

[2] Chen et al., "Targeted backdoor attacks on deep learning systems using data poisoning.

[3] Kurita et al., Weight Poisoning Attacks on Pre-trained Models.

[4] Carlini et al., Poisoning Web-Scale Training Datasets is Practical.

[5] Shi et al., Optimization-based prompt injection attack to llm-as-a-judge.

[6] Goodside, Prompt injection attacks against GPT-3, https://simonwillison.net/2022/Sep/12/prompt-injection/.

**Questions:**

1. The three threat models resemble existing paradigms in backdoor and poisoning literature. Could the authors clarify what new challenges or mechanisms specifically arise when these threats are instantiated in LLM agents rather than traditional models?
2. To what extent are the observed vulnerabilities caused by the agent’s unique architectural components (e.g., planner, memory, tool-use APIs) vs. inherent LLM weaknesses? Can we perform any ablation to isolate which modules are most responsible for these vulnerabilities?
3. The paper introduces “environmental poisoning,” which appears conceptually similar to prompt injection attacks already discussed in prior literature. Could the author explain how environmental poisoning differs mechanistically or conceptually from standard prompt injection? What are the unique contributions of the proposed attack compared to existing prompt injection attacks?

---

> ### Author Response · Authors · 2025-11-26
>
> Thank you. We would like to emphasize that there is a significant discrepancy between the content of this review and the content of our work. For example, AutoGPT, BabyAGI, and LangChain-based agents are not mentioned in the paper.
>
> Nevertheless, we address the reviewer’s concerns about novelty and ablations in the general response.
>
> Regarding environmental poisoning, we want to clarify that "Environmental poisoning" (TM2) is a data poisoning attack that occurs during trace collection (i.e., "prompt-injecting the teacher model"), thereby poisoning the data collected for fine-tuning. It is not a standard, inference-time prompt injection attack.

---

### Official Review · Reviewer_kfL9 · 2025-11-01

**Soundness:** 3
**Presentation:** 3
**Contribution:** 3
**Rating:** 6
**Confidence:** 3

**Summary:**

The paper shows that agentic AI systems trained via self-interaction fine‑tuning are vulnerable to supply‑chain backdoors (through training, data‑collection, or via base weights). The authors show on web and tool‑calling agents on WebArena and tau‑Bench, poisoning only 2-5% of traces yields high attack success rates (>80–100%). The backdoors persist through fine‑tuning.

**Strengths:**

+ The supply-chain framing with three threat models is nice

+ The finding that small poison ratios can yield very high attack success without interfering with task success appears to be an important finding

+ The experiment design is strong

**Weaknesses:**

- There's no side-by-side comparison with existing backdoor baselines, liek BadAgent or AgentPoison, which makes the novelty a bit harder to evaluate

- The triggers for WebArena seem to rely on very long passages. I think testing on shorter (more realistic) triggers would help

- There don't appear to be results for very low poison levels (under 2%)

- Fig. 3 shows steep gains but the sparsity of the plot makes it really hard to fairly evaluate

**Questions:**

- How resilient is the attack to variations in the trigger (namely length)?

- What is the smallest poison budget (either number of samples or tokens) that still produces a reliable activation? Can the authors provide confidence intervals for these thresholds?

- How do different adaptation methods compare (DPO, RLAIF, etc.) in terms of persistence of backdoors? Can any regime reduce ASR without a drop in task success? I'm struggling a bit to see what is special about the studied regimes.

---

> ### Author Response · Authors · 2025-11-26
>
> We thank the reviewer for the effort they put into reviewing our work. Their comments and suggestions significantly improve the work.
>
> **Ablations:** We ran the suggested experiments and report the results in the general response (trigger length, trigger type, lower poisoning levels, and even model sizes). Please let us know if you require further clarification.
>
> **Novelty:** We clarify the novelty aspects of our work in the general response, but we directly address the works you mention here. AgentPoison targets poisoning a RAG memory. We study supply-chain attacks on agentic models, and RAG memories are typically created in a user’s context rather than during model development, so this threat model does not apply to our setting. BadAgent is limited to TM1 and focuses on LoRA-based poisoning and defense via clean finetuning. In contrast, our work examines full finetuning, demonstrating that it remains ineffective as a defense (reinforcing their findings), and evaluates a broader set of defense strategies, including a novel approach.
>
> **Adaptation methods:** Excellent question! We answer this point in the general response.

---

### Official Review · Reviewer_RJ5c · 2025-11-03

**Soundness:** 2
**Presentation:** 2
**Contribution:** 1
**Rating:** 2
**Confidence:** 4

**Summary:**

This paper is an experiment report of backdoor attacks on 3 stages of deploying LLM agents. It shows that AI supply chain is vulnerable to backdoor attacks.

**Strengths:**

1. This paper provides experimental results of 3 threat models with various attack and defense settings.

**Weaknesses:**

1. This paper hasn't proposed new methods for backdoor attacks or defense for LLM agents.
2. The vulnerability of LLM agents in different scenarios has already been studied, as also discussed in Related Work.
3. The contribution of this work is limited, without giving new methds or results about backdoor attacks on LLM agents.

**Questions:**

Please reply to Weaknesses and clarify the novelty of this paper.

---

> ### Author Response · Authors · 2025-11-26
>
> We thank the reviewer for their time and effort in assessing our work. We think that this review overlooks some key aspects of the paper (e.g. the supply-chain framing, the novel environmental poisoning strategy and the novel use of defenses). Most of the criticisms relate to novelty, which we address in the general comment.

---

### Author Response · Authors · 2025-11-26
**General Comment**

We thank the reviewers for their constructive feedback. They highlighted several strengths of our work: (1) effectively identifying vulnerabilities in the agentic supply chain (RJ5c, kfL9, n4Sw, mhKC), with one calling these “critical and previously underexplored” (mhKC), and (2) exposing flaws in existing defense strategies (n4Sw). Reviewers also emphasized the quality, systematicity, and compelling nature of our experiments (kfL9, n4Sw, mhKC), the importance of our finding that very small poison rates still yield high attack success (kfL9), the generalizability of results across both web and tool-calling agents (n4Sw), and the comprehensive coverage of threat models and defenses (n4Sw). Below, we address common concerns about novelty and ablations.

## Novelty
We appreciate the reviewers’ request for greater clarity regarding novelty and outline below the contributions that uniquely distinguish our work.

**Supply chain framing:** This is the first work to study vulnerabilities in the agentic AI supply chain as a whole and across multiple domains (web and tool-calling agents), as noted by kfL9. With the rapid productization of agents, it is crucial for the community to understand what can go wrong and where threats emerge.

**Environment poisoning (TM2):** By grounding our analysis in current real-world practices, we introduce TM2 (environment poisoning), a threat model that has not, to our knowledge, been examined in the context of agentic systems. Although data poisoning has been explored for LLMs trained on internet-scale corpora [1], TM2 only becomes relevant for agents when practitioners collect unsupervised data directly from the wild, a practice adopted only very recently (for example, NetNav [2], AgentSynth [3], InSTA [4]). Our work is therefore the first to formalize and analyze this emerging vulnerability for agents.

**Small-poison backdoors (TM1–TM2):** While data poisoning (TM1) has been explored previously (e.g., BadAgent [5]), we provide a novel analysis demonstrating that very small poison ratios can reliably induce backdoors without compromising task success. This finding, highlighted by kfL9 and later reinforced by independent work from Anthropic [6], is new to the agentic literature.

**Exposing defense failures:** In contrast to prior work, we systematically evaluate a broad range of state-of-the-art defenses and demonstrate that they fail under realistic agentic settings for all threat models, including full finetuning (beyond LoRA, as in [5]), weight inspection, firewalls, and guardrails. Most of these had never been studied in this setting.

**A new defense direction:** Motivated by these failures, we introduce a simple but novel defense based on an LLM judge. Its effectiveness in the web setting and its failure in the tool-calling setting reveal a new open problem for the community: contextual guardrails for agentic systems.

We hope this clarifies the novelty and significance of our work and will revise the paper accordingly.

## Ablations
To strengthen our claims, we conducted a series of additional experiments and ablations.

### Trigger size (Web Setting)
We performed ablations on the number of tokens used in the trigger in the web setting and find consistent results across different poisoning levels.

| Trigger size (tokens) | Poisoning level (%) | TSR | ASR |
| :--- | :--- | :--- | :--- |
| **100** | **5** | 14.09 ± 1.9 | 98.2 ± 0.7 |
| **100** | **25** | 16.51 ± 2.0 | 100 ± 0 |
| **500** | **5** | 16.0 ± 2.0 | 90.9 ± 1.5 |
| **500** | **25** | 19.1 ± 2.2 | 100 ± 0 |


### Trigger type (Web Setting)

In addition to trigger size, we also consider ablations on trigger type in the web setting. Whereas the trigger we use in the paper is a series of tokens forming a coherent text on the importance of website accessibility ( “Website accessibility and AI agent design represent critical intersections of technology [...]” ), in this ablation, we consider a trigger made of seemingly random configuration strings (e.g. “request_id: req_2a5f8c1d-9e4b-4a7c-b3d6-8f1e5a9c2d7b, session: sess_4d7a1e9f-2c5b-4e8a-9f3d-6c2a8e5f1b7d, trace_id: trc_8b2f5d1a-6e9c-4f2a-8d5b-3a7e1c9f6d2b, […]”). We present results in the following table:

| Trigger size (tokens) | Poisoning level (%) | TSR (%) | ASR (%) |
| :--- | :--- | :--- | :--- |
| **1000** | **5** | 15.6 ± 2.0 | 0.6 ± 0.4 |
| **1000** | **25** | 13.6 ± 1.9 | 47.3 ± 2.7 |
| **1000** | **50** | 11.5 ± 1.8 | 97.0 ± 1.0 |

We find that using random triggers is detrimental to TSR. As the poisoning level increases, we observe that TSR diminishes up to a similar point as the zero-shot baseline of 11.1 ± 1.7 . Concurrently, we observe that ASR is much lower than it is with the base trigger, highlighting that learning an association between a seemingly random trigger and a malicious action is more complex.

---

> ### Author Response · Authors · 2025-11-26
> **General Response - continued**
>
> ### Smaller Poisoning levels (Tool-calling & web settings)
> We perform ablations on smaller poisoning percentages, with triggers exactly in line with the paper and report the numbers here:
>
> #### Web Setting
>
> | Poisoning level (%) | Number of poisoned samples | TSR (%) | ASR (%) |
> | :--- | :--- | :--- | :--- |
> | **1** | **454** | 16.51 ± 2.0 | 61.27 ± 2.7 |
> | **2** | **908** | 16.22 ± 2.0 | 68.77 ± 2.6 |
>
> #### Tool-calling Setting
>
> | Poisoning level (%) | Number of poisoned samples | TSR (%) | ASR (%) |
> | :--- | :--- | :--- | :--- |
> | **1** | **40** | 42.34 ± 2.57 | 0 |
> | **2** | **80** | 45.22 ± 2.68 | 1.02 ± 0.59 |
> | **2.5** | **100** | 39.71 ± 2.63 | 97.13 ± 1.0 |
> | **3** | **120** | 44.35 ± 2.67 | 98.96 ± 0.6 |
>
> In both settings, we find that lower poisoning rates can still plant backdoors in agentic models, although not with the same reliability as higher poisoning rates. Interestingly, for the tool-calling setting, we observe that using as little as 100 poisoned samples leads to a highly effective backdoor, echoing concurrent work [6]. Still in this setting, we manage to find that using 40 samples is not enough to plant a backdoor, as malicious behaviors emerge from 80 samples. In the web setting, we have not found this breaking point yet, but we are inclined to conduct further experiments and report them in the camera-ready version.
>
> ### Model size (tool-calling setting)
>
> As requested, we perform additional experiments to highlight that attacks are still effective with larger models. This again echoes concurrent work, showing that backdoors planted with a few samples can poison model across sizes [6].
>
> | Poisoning level (%) | Number of samples | Model size | TSR (%) | ASR (%) |
> | :--- | :--- | :--- | :--- | :--- |
> | **5** | **200** | **14B** | 41.27 ± 2.70 | 99.64 ± 0.36 |
>
> ## Absence of RL-finetuning
> Adaptation methods: This is an excellent question. Agents are often trained with reinforcement learning (RL), but SFT typically bootstraps the initial policy [7]. Since we show that even minimal poisoned data in SFT induces an effective backdoor, the agent would likely acquire the backdoor before RL begins. RL rewards usually target task completion rather than intermediate actions, so a backdoored behavior that still completes the task may be reinforced. Process supervision, for example through process reward models, may help mitigate this by supervising intermediate steps. We will add this discussion to the paper and would be happy to explore the effect of RL as a mitigation in the camera-ready, although we cannot run additional experiments within the rebuttal timeline.
>
>
> [1] https://arxiv.org/abs/2302.10149
>
> [2] https://arxiv.org/abs/2410.02907
>
> [3] https://arxiv.org/abs/2506.14205
>
> [4] https://arxiv.org/abs/2502.06776
>
> [5] https://arxiv.org/abs/2406.03007
>
> [6] https://www.anthropic.com/research/small-samples-poison
>
> [7] https://arxiv.org/abs/2507.04103

---

> ### Author Response · Authors · 2025-11-28
>
> We have updated the paper according to reviewers' comments, with updates displayed in blue.

---

### Meta-Review · Area_Chair_2WJo · 2025-12-15

**Summary:**

This paper tackled backdoor-style attacks in the AI agent supply chain and highlighted a practically important, underexplored vulnerability. Reviewers appreciated the problem framing and generally solid experimental design, but raised highly consistent concerns: (a) the conceptual novelty over existing poisoning and prompt-injection work remains only partially articulated; (b) the empirical evaluation lacks comparisons to strong baselines, larger models, and more diverse, realistic benchmarks; (c) agent-specific questions, including component-level ablations, adaptation and defense strategies, broader agent architectures, are not systematically studied. The overall evidences are still quite limited for acceptance, and thus I recommend reject.

**PS, the authors should not include the acknowledgements with many real names before the camera-ready version.**

**Reviewer Concerns:**

**RJ5c** Concerns: limited conceptual novelty; no clearly new attack or defense beyond existing poisoning/backdoor work; incremental empirical contribution. Addressed: the environmental/supply-chain poisoning threat model. Outstanding: unknown strength of conceptual advance and distinctiveness over existing backdoor/poisoning literature.

**kfL9** Concerns: missing comparisons to strong backdoor/poisoning baselines; unrealistically long/visible triggers; high poison rates; limited analysis of agent adaptation mechanisms. Addressed: additional ablations on trigger size/type, poison rate, and model size. Outstanding: no direct comparisons to strong baselines and systematic adaptation/defense studies.

**n4Sw** Concerns: novelty relative to prompt-injection and data-poisoning literature; lack of agent-specific/component-level ablations; unclear generality beyond studied setups.
Addressed: environmental poisoning vs. prompt injection. Outstanding: no individual components and broader agent settings.

**mhKC** Concerns: limited scale (models and datasets), narrow benchmark suite, and absence of stronger baselines/ablations. Addressed: some added experiments and modest scaling. Outstanding: larger models, richer benchmarks, and deeper ablation/robustness analyses still needed.

**Reviewer Scores:**

I don't think any of the four reviewers would like to increase his or her score.

---

### Decision · Program_Chairs · 2026-01-26

Reject